# Three-Stage Operational Amplifier with Frequency Compensation Using Cascade Zero

**Yurin Jin** , **Yeonho Seo, Sungmi Kim and Seongik Cho \***

Department of Electronics Engineering, Jeonbuk National University, Jeonju 54896, Republic of Korea; 201650164@jbnu.ac.kr (Y.J.); kiru8@keti.re.kr (Y.S.)
* Correspondence: sicho@jbnu.ac.kr

**Abstract:** Short channel MOSFET exhibits the characteristics of wide bandwidth and low DC gain. A low DC gain causes a high gain error and a narrow output linear range in the closed loop. The DC gains can be improved by using the cascade structure, but frequency compensation is required due to the increase in the number of poles. The output nodes of each stage in a cascade Common-Source amplifier have a cascade of zero, and this zero is cancelled out by the input node of the next stage. This paper proposes a three-stage operational amplifier (op-amp) with frequency compensation using cascade zero. This op-amp was implemented in the 180 nm CMOS technology and achieved 86.96 MHz unity–gain frequency, 51.7° phase margin at 32 pF load capacitor and 99.83 dB DC gain, that is, a 36.21 dB improvement over a two-stage op-amp with the same power consumption. The op-amp consumed 7.74 mW with a supply voltage of 1.8 V.

**Keywords:** op-amp; gain boost; frequency compensation

## 1. Introduction

Short channel MOSFET exhibits the advantage of wide bandwidth because of low parasitic capacitors and channel resistance, but the DC gain is low due to the low output resistance [1,2]. A low DC gain causes a high gain error and a narrow output linear range in the closed loop [3,4]. This problem can be improved by using cascode structures, cascade structures, gain boosting circuits or error correction circuits [4,5]. The DC gain of a cascade amplifier can be increased by adding more amplifier stages, but frequency compensation is necessary. This is because the number of poles increases with the number of stages added [6]. Figure 1 shows the two-stage Common-Source (CS) amplifier.

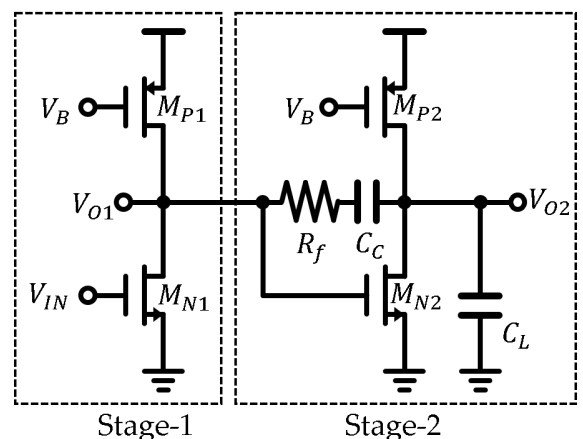

**Figure 1.** Schematic of two-stage CS amplifier.

This circuit consists of two CS amplifiers and a compensation capacitor ($C_C$). $V_B$ is bias voltage. The amplifier exhibits two poles at the $V_{O1}$ and $V_{O2}$ nodes. By using the $C_C$,

the pole frequency at the $V_{O1}$ node can be lowered, resulting in a decrease in the unity–gain frequency. The small signal equivalent circuit for ac analysis is shown in Figure 2.

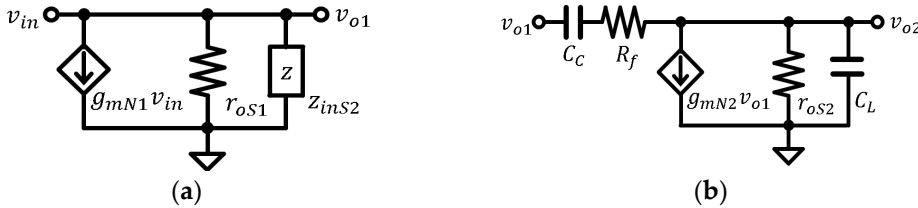

**Figure 2.** The small signal equivalent circuit of the two-stage CS amplifier. (**a**) Small signal model of Stage 1. (**b**) Small signal model of the first Stage 2.

The $r_{oS1}$ and $r_{oSs}$ are the output resistance of each stage, and $z_{inS2}$ is the input impedance of Stage 2. These are expressed as:

$$r_{oS1} = (r_{oN1} \parallel r_{oP1}) \tag{1}$$

$$r_{oS2} = (r_{oN2} \parallel r_{oP2}) \tag{2}$$

$$z_{inS2}(s) = \frac{1 + r_{oS2}C_C s}{(1 + r_{oS2}g_{mN2})C_C s}, \quad \left(s \ll \frac{1}{r_{oS2}C_L}\right) \text{ and } \left(s \ll \frac{1}{R_f C_C}\right) \tag{3}$$

$r_{oN}$ and $r_{oP}$ are the output resistance of NMOS and PMOS, and $g_{mN}$ is the transconductance of NMOS. The cascade zero ($z_C$) and the gains of each stage are expressed as:

$$z_C = \frac{1}{r_{oS2}C_C} \tag{4}$$

$$
\begin{aligned}
\frac{V_{O1}(s)}{V_{IN}(s)} &= -g_{mN1}(r_{oS1} \parallel z_{inS2}(s)) = -\frac{g_{mN1}r_{oS1}\left(1 + \frac{1}{z_C}s\right)}{1 + (r_{oS1} + r_{oS2} + r_{oS1}r_{oS2}g_{mN2})C_C s} \\
&\approx -\frac{g_{mN1}r_{oS1}\left(1 + \frac{1}{z_C}s\right)}{1 + r_{oS1}r_{oS2}g_{mN2}C_C s}, \quad (r_{oS1}r_{oS2}g_{mN2} \gg r_{oS1} + r_{oS2})
\end{aligned}
\tag{5}
$$

$$\frac{V_{O2}(s)}{V_{O1}(s)} = -\frac{g_{mN2}r_{oS2}\left(1 - \left(\frac{1}{g_{mN2}} - R_f\right)C_C s\right)}{\left(1 + \frac{1}{z_C}s\right)\left(1 + \frac{1}{g_{mN2}}C_L s\right)} \tag{6}$$

Equation (5) reveals that the output nodes of each stage contain a cascade zero that cannot be analyzed using the Miller effect, but this zero is cancelled out by the input node of the next stage. The frequency response, DC gain ($A_{DC,2ST}$) and unity–gain frequency ($\omega_{GBW,2ST}$) of this amplifier are expressed as:

$$\frac{V_{O2}(s)}{V_{IN}(s)} = \frac{g_{mN1}r_{oS1}g_{mN2}r_{oS2}\left(1 - \left(\frac{1}{g_{mN2}} - R_f\right)C_C s\right)}{(1 + r_{oS1}r_{oS2}g_{mN2}C_C s)\left(1 + \frac{1}{g_{mN2}}C_L s\right)} \tag{7}$$

$$A_{DC,2ST} = \frac{V_{O2}(0)}{V_{IN}(0)} = g_{mN1}r_{oS1}g_{mN2}r_{oS2} \tag{8}$$

$$\omega_{GBW,2ST} = \frac{g_{mN1}r_{oS1}g_{mN2}r_{oS2}}{r_{oS1}r_{oS2}g_{mN2}C_C} = \frac{g_{mN1}}{C_C}, \quad \left(\frac{1}{g_{mN3}} = R_f\right) \tag{9}$$

Equation (7) shows that the cascade zero in Equation (5) is cancelled out by the pole in Equation (6). This paper proposes a three-stage operational amplifier (op-amp) with frequency compensation using cascade zero that is designed based on the two-stage

op-amp. For a performance comparison, the conventional two-stage op-amp and the proposed op-amp were implemented on a single chip using 180 nm CMOS technology. The measurement results indicate that the proposed op-amp had the same power consumption, phase margin and unity–gain frequency as the two-stage op-amp, while offering a higher gain. This paper is organized as follows. Sections 2 and 3 explain the compensation method and proposed op-amp. Sections 4 and 5 show the simulation and measurement result, respectively. Section 6 concludes this paper.

## 2. Compensation Method

Figure 3 shows the three-stage CS amplifier with frequency compensation using cascade zero. The sizes of the MOSFET in Stage 2.1 and Stage 2.2 are the same. In this method, the phase margin is compensated by canceling the pole of Stage 3 with a cascade zero, allowing for an increase in the DC gain without the degradation of the unity–gain frequency.

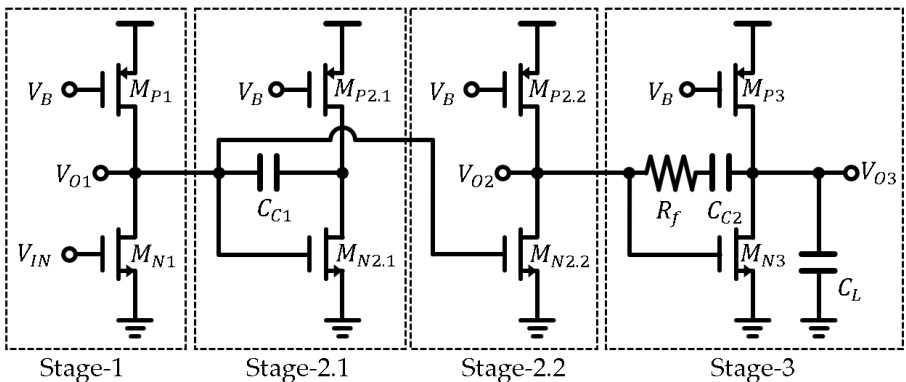

Stage-1      Stage-2.1      Stage-2.2      Stage-3

**Figure 3.** Schematic of the three-stage CS amplifier with frequency compensation using cascade zero.

$r_{oS3}$ is the output resistance of Stage 3. The cascade zeros ($z_{C1}$, $z_{C2}$), $r_{oS3}$ and the gains between $V_{IN}$ and the output node of each stage and DC gain ($A_{DC,3ST}$) are expressed as:

$$r_{oS3} = (r_{oN3} \parallel r_{oP3}) \tag{10}$$

$$z_{C1} = \frac{1}{r_{oS2}C_{C1}} \tag{11}$$

$$z_{C2} = \frac{1}{r_{oS3}C_{C2}} \tag{12}$$

$$\frac{V_{O1}(s)}{V_{IN}(s)} = -\frac{g_{mN1}r_{oS1}\left(1 + \frac{1}{z_{C1}}s\right)}{1 + r_{oS1}r_{oS2}g_{mN2}C_C s} \tag{13}$$

$$\frac{V_{O2}(s)}{V_{IN}(s)} = \frac{g_{mN1}r_{oS1}g_{mN2}r_{oS2}\left(1 + \frac{1}{z_{c1}}s\right)\left(1 + \frac{1}{z_{c2}}s\right)}{(1 + r_{oS1}r_{oS2}g_{mN2}C_{C1}s)(1 + r_{oS2}r_{oS3}g_{mN3}C_{C2}s)} \tag{14}$$

$$\frac{V_{O3}(s)}{V_{IN}(s)} = -\frac{g_{mN1}r_{oS1}g_{mN2}r_{oS2}g_{mN3}r_{oS3}\left(1 + \frac{1}{z_{c1}}s\right)\left(1 - \left(\frac{1}{g_{mN3}} - R_f\right)C_{C2}s\right)}{(1 + r_{oS1}r_{oS2}g_{mN2}C_{C1}s)(1 + r_{oS2}r_{oS3}g_{mN3}C_{C2}s)\left(1 + \frac{1}{g_{mN3}}C_L s\right)} \tag{15}$$

$$A_{DC,3ST} = \frac{V_{O3}(0)}{V_{IN}(0)} = g_{mN1}r_{oS1}g_{mN2}r_{oS2}g_{mN3}r_{oS3} \tag{16}$$

$z_{C1}$ in Equation (13) is added by the input impedance of Stage 2.1, and $V_{O1}$ is amplified by Stage 2.2 to prevent the elimination of the $z_{C1}$. In Equation (15), the $z_{C2}$ in Equation (14) is cancelled out by Stage 3, as in Equation (7), but $z_{C1}$ is not cancelled. The poles ($p_1$, $p_2$, $p_3$) of the circuit are expressed as follows:

$$p_1 = \frac{1}{r_{oS1} r_{oS2} g_{mN2} C_{C1}} \tag{17}$$

$$p_2 = \frac{1}{r_{oS2} r_{oS3} g_{mN3} C_{C2}} \tag{18}$$

$$p_3 = \frac{g_{mN3}}{C_L} \tag{19}$$

The $p_1$ is added by Stage 2.1. The unity–gain frequency ($\omega_{GBW,3ST}$) and rate of change of unity–gain frequency due to Stage 2.1 and Stage 2.2 ($\omega_{GBW,change}$) are calculated as:

$$\omega_{GBW,3ST} = \frac{A_{DC,3ST} p_1 p_2}{z_{C1}} = \frac{g_{mN1}}{C_{C2}}, \quad \left( \frac{1}{g_{mN3}} = R_f \right) \tag{20}$$

$$\omega_{GBW,change} = g_{mN2} r_{oS2} \times \frac{p_1}{z_{C1}} \times \frac{r_{oS2}}{r_{oS1}} = 1 \tag{21}$$

Equation (21) shows that the unity–gain frequency is not changed by Stage 2.1 and Stage 2.2. Figure 4 shows the frequency response based on $z_{C1}$, $p_1$, $p_2$, $p_3$ and $A_{DC,3ST}$. Figure 4 shows that the two-stage CS amplifier and the compensated three-stage CS amplifier have the same phase margin and unity–gain frequency.

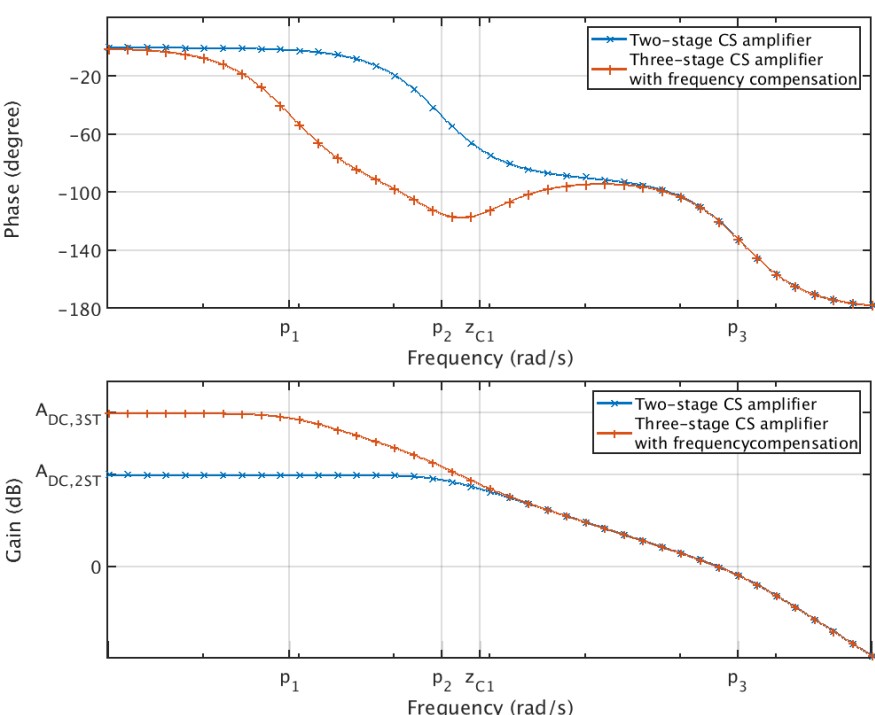

**Figure 4.** Frequency response based on $z_{C1}$, $p_1$, $p_2$, $p_3$ and $A_{DC,3ST}$.

## 3. Proposed op-amp

### 3.1. Conventional Two-Stage op-amp

The proposed op-amp is designed by applying these methods to a conventional two-stage op-amp. Figure 5 shows the conventional two-stage op-amp [7,8].

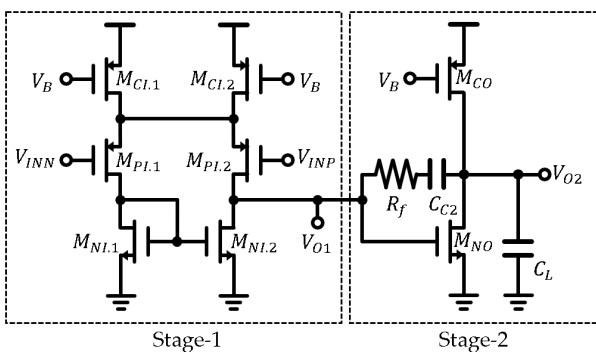

**Figure 5.** Schematic of a conventional two-stage op-amp.

The output resistance of each stage ($r_{oI}$, $r_{oO}$), frequency response ($A_{2ST-opamp}$), DC gain ($A_{DC,2ST-opamp}$) and unity–gain frequency ($\omega_{GBW,2ST-opamp}$) of the two-stage op-amp can be expressed as:

$$r_{oI} = (r_{oIN} \parallel r_{oIP}) \tag{22}$$

$$r_{oO} = (r_{oON} \parallel r_{oOP}) \tag{23}$$

$$A_{2ST-opamp} = \frac{V_{O2}(s)}{V_{INP}(s)} = \frac{g_{mPI}r_{oI}g_{mNO}r_{oO}\left(1 - \left(\frac{1}{g_{mNO}} - R_f\right)C_{C2}s\right)}{(1 + sg_{mNO}r_{oI}r_{oO}C_{C2}s)\left(1 + \frac{1}{g_{mNO}}C_L s\right)} \tag{24}$$

$$A_{DC,2ST-opamp} = \frac{V_{O2}(0)}{V_{IN}(0)} = g_{mPI}r_{oI}g_{mNO}r_{oO} \tag{25}$$

$$\omega_{GBW,2ST-opamp} = \frac{g_{mPI}}{C_{C2}} \tag{26}$$

### 3.2. Proposed op-amp

Figure 6 shows the three-stage op-amp with frequency compensation using cascade zero. The sizes of the MOSFET in Stage 1 and Stage 2 are the same. The second stage in Figure 3 is divided into Stage 2.1 for compensation and Stage 2.2 for amplification. However, Stage 2 in Figure 6 has differential inputs, so compensation and amplification are operated in one stage.

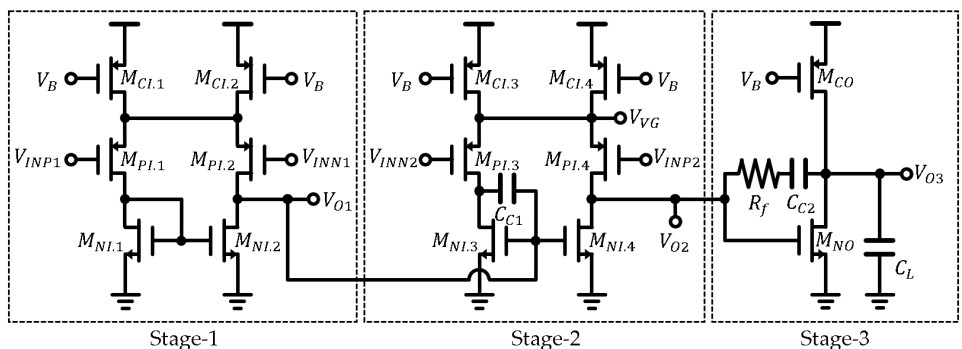

**Figure 6.** Schematic of three-stage op-amp with frequency compensation using cascade zero.

The frequency response and unity–gain frequency between $V_{INP2}$ and $V_{O3}$ nodes is equal to Equations (24) and (26). Since $M_{PI1.3}$ and $M_{PI1.4}$ have differential inputs ($V_{INP2}$, $V_{INN2}$), the $V_{VG}$ node is analyzed as a virtual. However, since $M_{NI1.3}$ and $M_{NI1.4}$ have the same input, $M_{CI1.3}$ and $M_{CI1.4}$ are analyzed as the cascade structure with respect to

the $V_{O1}$ node. The output resistance of Stage 2 with respect to the $V_{O1}$ node ($r_{oS2-VO1}$) is expressed as:

$$r_{oS2-VO1} = r_{oIN} \parallel (g_{mIP}r_{oIP}r_{oIC} + r_{oIP} + r_{oIC}) \approx r_{oIN}, \ (r_{oIN} \ll g_{mIP}r_{oIP}r_{oIC}) \qquad (27)$$

Equations (22) and (27) show that $r_{oS2-VO1}$ is greater than $r_{oI}$, which increases the DC gain. The cascade zeros ($z_{C1}, z_{C2}$) and the gains between $V_{INP1}$ and the output node of each stage are expressed as:

$$z_{C1} = \frac{1}{r_{oS2-VO1}C_{C1}} = \frac{1}{r_{oIN}C_{C1}} \qquad (28)$$

$$z_{C2} = \frac{1}{r_{oO}C_{C2}} \qquad (29)$$

$$\frac{V_{O1}(s)}{V_{INP1}(s)} = -\frac{g_{mPI}r_{oI}\left(1 + \frac{1}{z_{C1}}s\right)}{1 + r_{oI}r_{oIN}g_{mNI}C_{C1}s} \qquad (30)$$

$$\frac{V_{O2}(s)}{V_{INP1}(s)} = -\frac{g_{mPI}r_{oI}g_{mNI}r_{oIN}\left(1 + \frac{1}{z_{c1}}s\right)\left(1 + \frac{1}{z_{c2}}s\right)}{(1 + r_{oI}r_{oIN}g_{mNI}C_{C1}s)(1 + r_{oIN}r_{oON}g_{mNO}C_{C2}s)} \qquad (31)$$

$$\frac{V_{O3}(s)}{V_{INP1}(s)} = \frac{g_{mPI}r_{oI}g_{mNI}r_{oIN}g_{mNO}r_{oO}\left(1 + \frac{1}{z_{c1}}s\right)\left(1 - \left(\frac{1}{g_{mNO}} - R_f\right)C_{C2}s\right)}{(1 + r_{oI}r_{oIN}g_{mNI}C_{C1}s)(1 + r_{oIN}r_{oON}g_{mNO}C_{C2}s)\left(1 + \frac{1}{g_{mNO}}C_Ls\right)} \qquad (32)$$

Equations (31) and (32) show the gain improvement due to $r_{oS2-VO1}$ and compensation term due to cascade zero. The DC gain and unity–gain frequency of Equation (32) ($A_{DC,EQ32}$, $\omega_{GBW,EQ32}$) are expressed as:

$$A_{DC,EQ32} = \frac{V_{O3}(0)}{V_{INP1}(0)} = g_{mPI}r_{oI}g_{mNI}r_{oIN}g_{mNO}r_{oO} \qquad (33)$$

$$\omega_{GBW,EQ32} = \frac{g_{mPI}r_{oI}g_{mNI}r_{oIN}g_{mNO}r_{oO}(r_{oIN}C_{C1})}{(r_{oI}r_{oIN}g_{mNI}C_{C1})(r_{oIN}r_{oON}g_{mNO}C_{C2})} = \frac{g_{mPI}}{C_{C2}} \qquad (34)$$

When $V_{INP1}$ and $V_{INP2}$ are equal, The DC gain and unity–gain frequency of proposed amplifier ($A_{DC,3ST-opamp}$, $\omega_{GBW,3ST-opamp}$) are expressed as:

$$\begin{aligned} A_{DC,3ST-opamp} &= \frac{V_{O3}(0)}{V_{INP1}(0)} + \frac{V_{O3}(0)}{V_{INP2}(0)} = A_{DC,2ST-opamp} + A_{DC,EQ32} \\ &= A_{DC,2ST-opamp}(1 + g_{mNI}r_{oIN}) \end{aligned} \qquad (35)$$

$$\begin{aligned} \omega_{GBW,3ST-opamp} &= \omega_{GBW,2ST-opamp} + \omega_{GBW,EQ32} = 2\frac{g_{mPI}}{C_{C2}} \\ &= 2\omega_{DC,2ST-opamp} \end{aligned} \qquad (36)$$

The product of the transconductance and output resistance of NMOS and PMOS are calculated as:

$$r_{oN}g_{mN} = \sqrt{\frac{2\mu_n C_{OX}}{\lambda_N L_N}\frac{W_N}{|I_{DS}|}} \qquad (37)$$

$$r_{oP}g_{mP} = \sqrt{\frac{2\mu_p C_{OX}}{\lambda_P L_P}\frac{W_P}{|I_{DS}|}} \qquad (38)$$

$W_N$, $W_P$, $L_N$ and $L_P$ represent the width and length of the NMOS and PMOS, while $\lambda_N$ and $\lambda_P$ are channel-length modulation coefficients. $\mu_n$, $\mu_p$ and $C_{OX}$ are the mobility and oxide capacitance, and $I_{DS}$ is the drain-to-source current. Equations (37) and (38) show that the DC gain of the amplifier remains constant when both $I_{DS}$ and the MOSFET width

change at the same rate. If the drain-source current and the MOSFET width of Stage 1 and Stage 2 are halved, the unity–gain frequency ($\omega_{GBW,3ST-opamp,smae}$) can be expressed as:

$$\omega_{GBW,3ST-opamp,smae} = \frac{g_{mPI}}{C_{C2}} \tag{39}$$

Equations (37)–(39) show that the proposed op-amp has the same unity–gain frequency and improved DC gain at the same power consumption compared to the conventional two-stage op-amp. The capacitance of $C_{C1}$ is determined by the minimum phase between $z_{C1}$ and the second pole, which is calculated as:

$$-90 - arctan\sqrt{\frac{r_{oON}g_{mNO}C_{C2}}{C_{C1}}} + arctan\sqrt{\frac{C_{C1}}{r_{oON}g_{mNO}C_{C2}}} \tag{40}$$

The slew-rate ($SR$) of the two-stage op-amp and proposed op-amp is calculated as Equation (41) [9]. Since the $I_{DS,MCI.3}$ of the proposed op-amp is half that of the two-stage op-amp, when the load capacitance is small, the slew-rate is reduced by half. $C_{C1}$ does not affect the slew-rate. When $M_{PI.3}$ is on, the swing size decreases by the gain of Stage 3, while when $M_{PI.3}$ is off, it does not affect Stage 3.

$$SR = min\left( \frac{I_{DS,MCI.3} + I_{DS,MCI.4}}{C_{C2}}, \frac{I_{DS,MCO}}{C_{C2} + C_L} \right) \tag{41}$$

## 4. Simulation Results

In this paper, both the conventional two-stage op-amp and the proposed op-amp were designed on a single chip for performance comparison. AC, DC, and total harmonic distortion (THD) analysis were performed to confirm their performance. The effect of the mismatch was analyzed through Monte Carlo simulations. Additionally, the relationship between the phase margin and load capacitance was simulated. To ensure a fast settling response, a load capacitance smaller than the capacitance with a phase margin of 45° should be used [10]. The proposed op-amp, MOSFET width and current of Stage 1 and Stage 2 were designed to be half that of the two-stage op-amp. The MOSFET size and characteristics are shown in Tables 1 and 2.

**Table 1.** MOSFET size of proposed op-amp and two-stage op-amp.

| MOSFET | Two-Stage op-amp | | Proposed op-amp | |
|---|---|---|---|---|
| | Width (μm) | Length (μm) | Width (μm) | Length (μm) |
| $M_{CI}$ | 40 | 0.3 | 20 | 0.3 |
| $M_{PI}$ | 40 | 0.3 | 20 | 0.3 |
| $M_{NI}$ | 12 | 0.3 | 6 | 0.3 |
| $M_{CO}$ | 800 | 0.3 | 800 | 0.3 |
| $M_{NO}$ | 240 | 0.3 | 240 | 0.3 |

**Table 2.** Characteristics of proposed op-amp and two-stage op-amp.

| Parameter | Two-Stage op-amp | Proposed op-amp |
|---|---|---|
| Supply voltage (V) | 1.8 | 1.8 |
| Current consumption (mA) | 4.252 | 4.252 |
| $R_f$ (Ω) | 154.1 | 154.1 |
| $C_L$ (pF) | 32 | 32 |
| $C_{C1}$ (pF) | - | 9.6 |
| $C_{C2}$ (pF) | 2.41 | 2.41 |
| DC Gain (dB) | 63.59 | 99.19 |
| Unity–gain Frequency (MHz) | 82.70 | 81.03 |
| Phase margin (°) | 66.53 | 65.17 |

### 4.1. AC and DC Simulation

Figure 7 shows the simulated frequency response of the two-stage op-amp and the proposed op-amp. As a simulation result, the DC gain improved by 35.6 dB over the two-stage op-amp, and the phase margin and unity–gain frequency were the same.

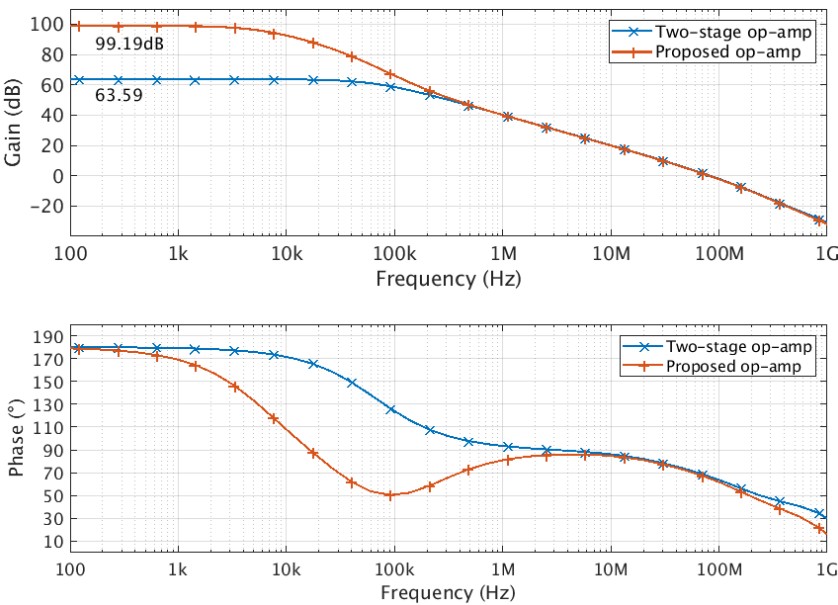

**Figure 7.** Simulated frequency response of the two-stage op-amp and the proposed op-amp.

To compare the operating range of the two op-amps, the DC gains according to the common mode input voltage and output voltage are shown in Figures 8 and 9. The DC gain of the proposed amplifier was higher than that of the conventional two-stage op-amp within the operation range of the op-amp, indicating that the operating range was not reduced.

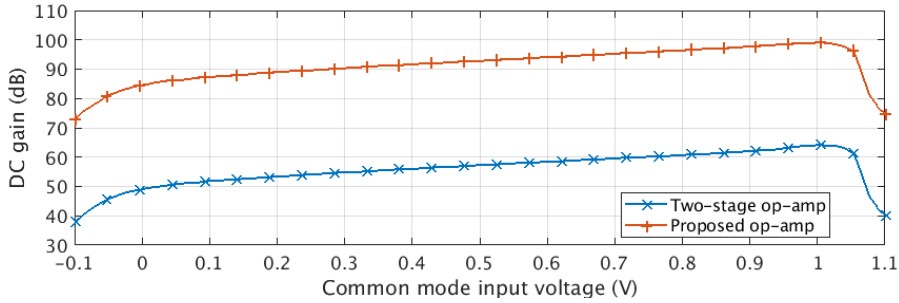

**Figure 8.** DC gain according to the common mode input voltage.

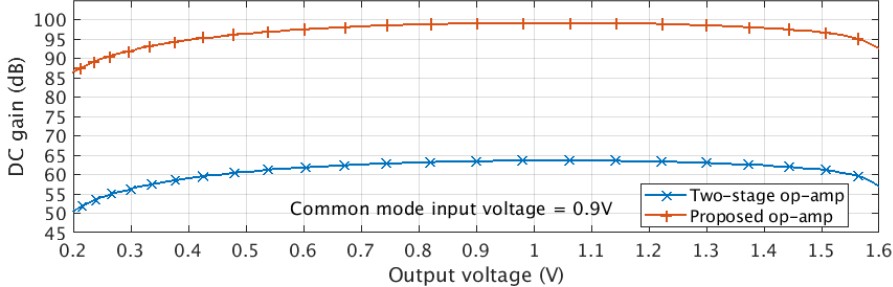

**Figure 9.** DC gain according to the output voltage.

### 4.2. THD Analysis Simulation

Figure 10 shows the circuit used for THD simulation in a closed loop. $V_{IN}$ is a sine wave with an amplitude of 0.7 V, a frequency of 1 kHz and centered on $V_{CM}$, which is 0.9 V. The THD simulation results are shown in Figures 11 and 12. The simulation results show that the THD of the closed loop improved by 35.14 dB due to the increase in DC gain.

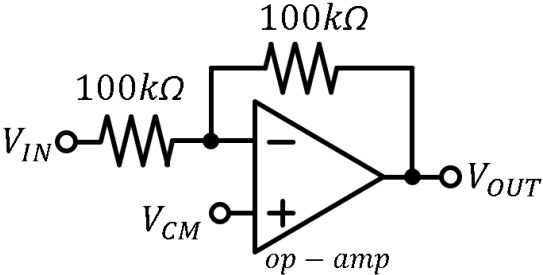

**Figure 10.** The circuit for THD simulation in a closed loop.

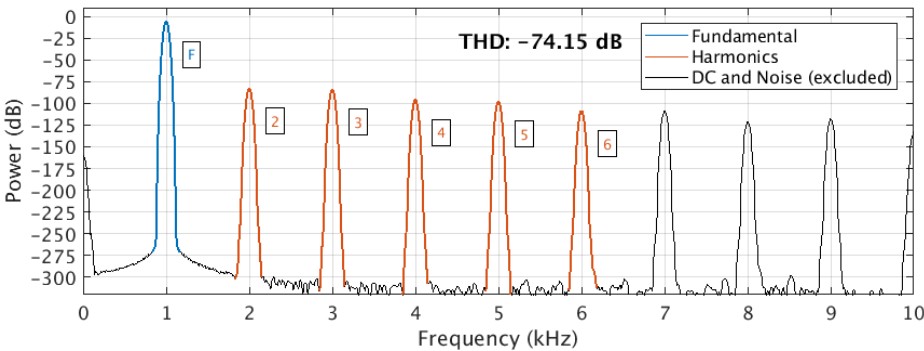

**Figure 11.** Simulated THD of the two-stage op-amp.

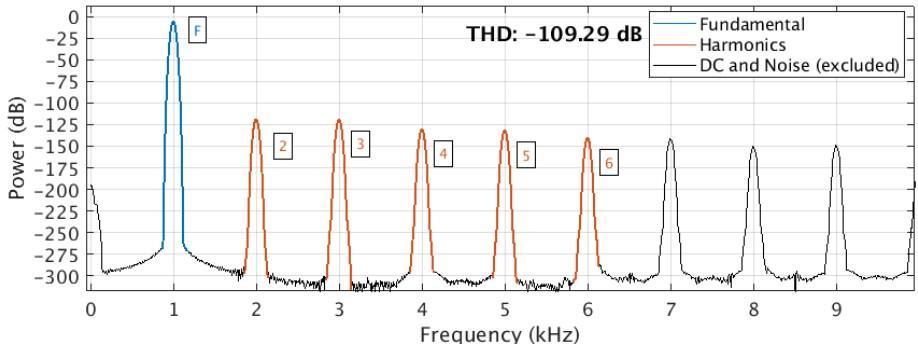

**Figure 12.** Simulated THD of the proposed op-amp.

### 4.3. Monte Carlo Simulation

A Monte Carlo simulation was performed to simulate the performance changes due to mismatch, and the results are presented in Figures 13–15. Simulations were performed 100 times each for the DC gain, phase margin and unity–gain frequency of the proposed op-amp and the two-stage op-amp. The results show that the proposed op-amp operated normally within the mismatch range of the process.

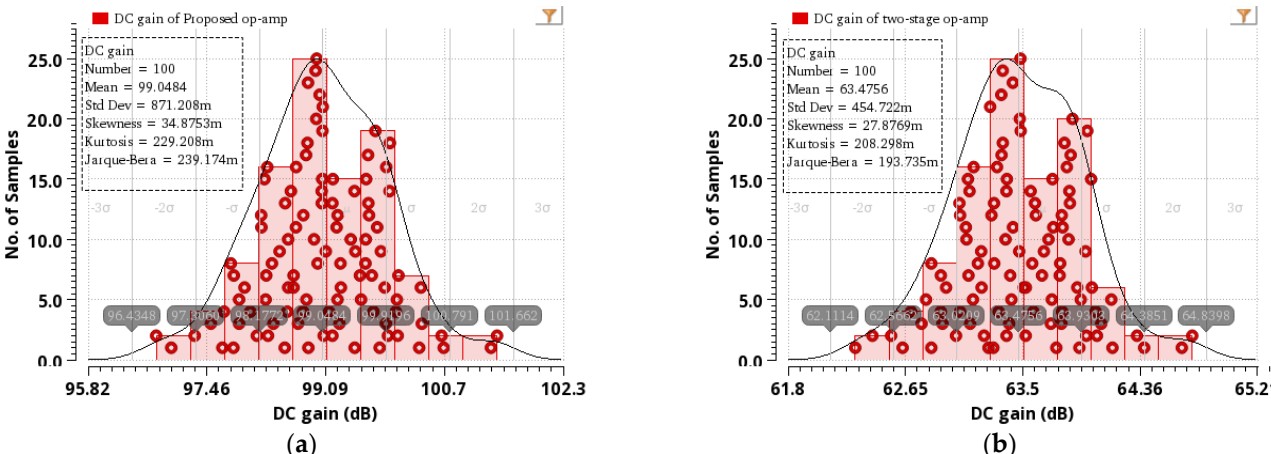

**Figure 13.** Monte Carlo simulation results of DC gain: (**a**) result of the proposed op-amp; (**b**) result of the two-stage op-amp.

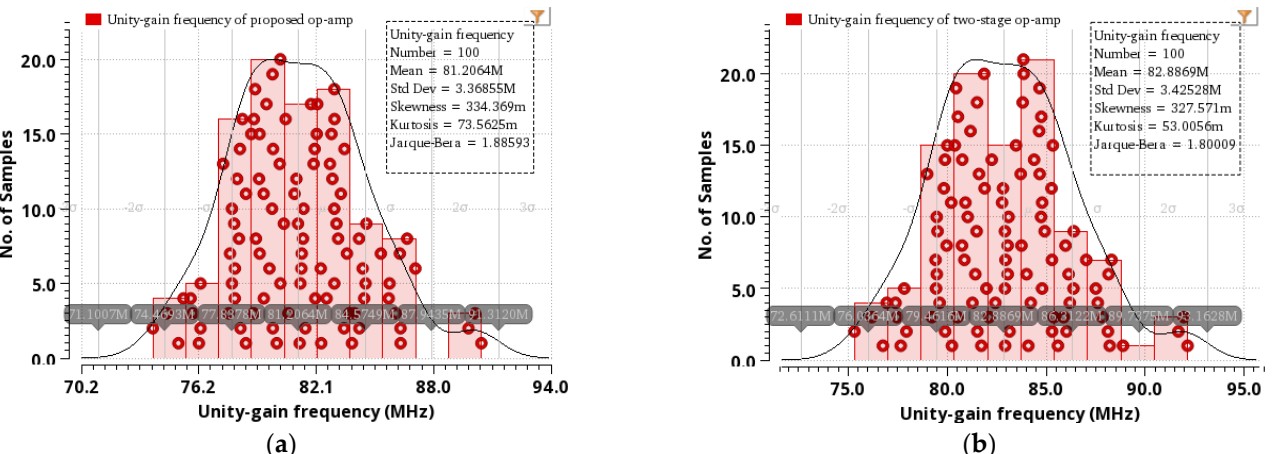

**Figure 14.** Monte Carlo simulation results of unity-gain frequency: (**a**) result of the proposed op-amp; (**b**) result of the two-stage op-amp.

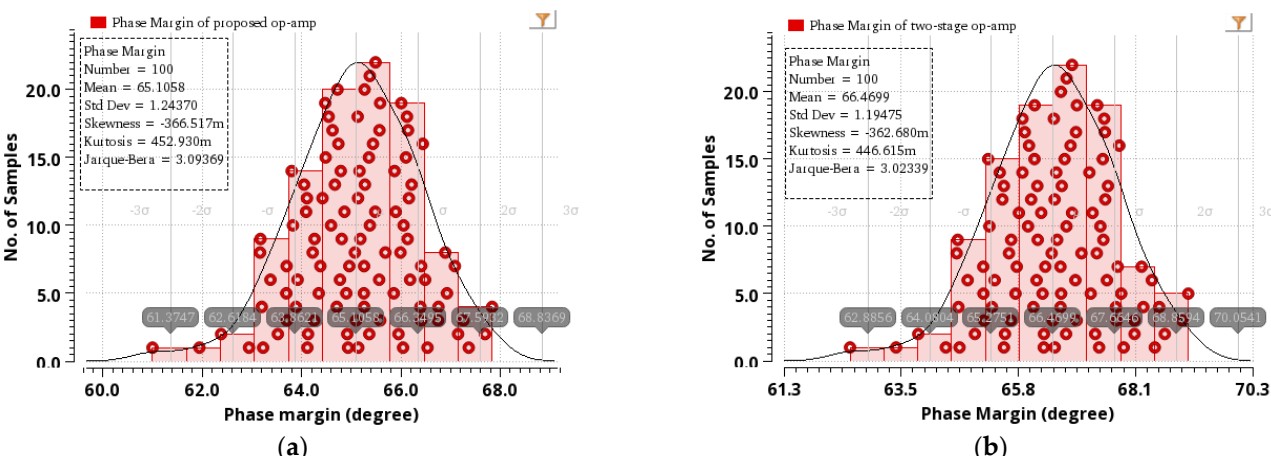

**Figure 15.** Monte Carlo simulation results of phase margin: (**a**) result of the proposed op-amp; (**b**) result of the two-stage op-amp.

### 4.4. The Relationship between Phase Margin and Load Capacitance

Figure 16 shows the relationship between the phase margin and the load capacitance. The proposed op-amp and the two-stage op-amp had phase margins of 45° with a load capacitance of 90.82 pF and 97.39 pF, respectively.

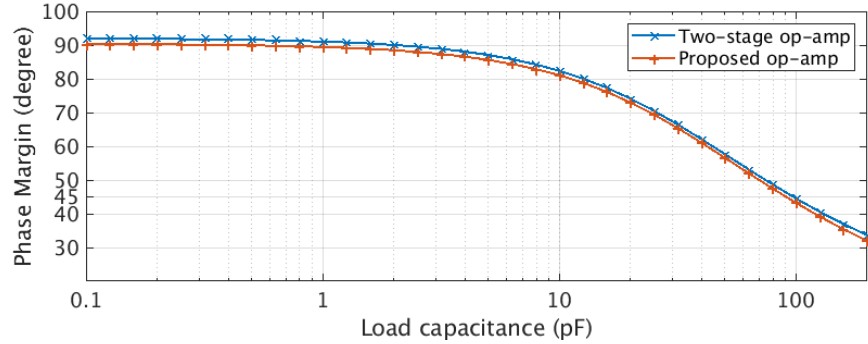

**Figure 16.** The relationship between phase margin and load capacitance of the proposed op-amp and two-stage op-amp.

### 5. Measurement Results

Figure 17 shows the layout and photograph of the proposed op-amp and two-stage op-amp implemented in the 180 nm CMOS technology. Section A shows the proposed op-amp, 146 μm × 102 μm, and section B shows the conventional two-stage op-amp, 96 μm × 70 μm.

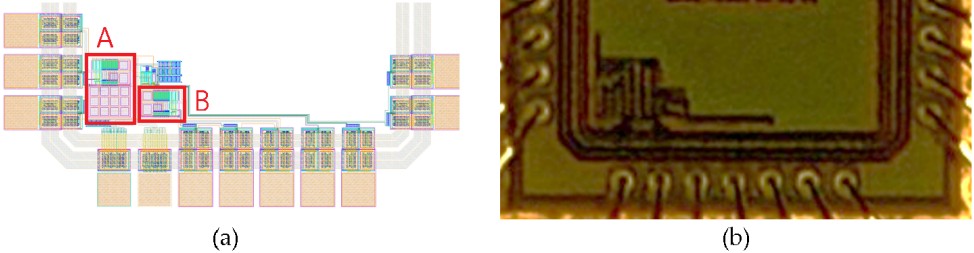

(a) (b)

**Figure 17.** (**a**) Layout of the proposed circuit; (**b**) photograph of the proposed circuit.

### 5.1. DC Gain Measurement

Figure 18 shows the DC gain measurement circuit for the two-stage op-amp [10]. The input voltage ($V_{IN}$) changed from −0.3 V to 0.3 V at 50 mV intervals, and the output voltage ($V_{OUT}$) was measured at 1024 Sample/s for 100 s. The DC gain of the op-amp ($A_{DC}$) is calculated as:

$$A_{DC} = -\frac{R_2 + R_3}{R_2}\frac{\Delta V_{IN}}{\Delta V_{OUT}} \tag{42}$$

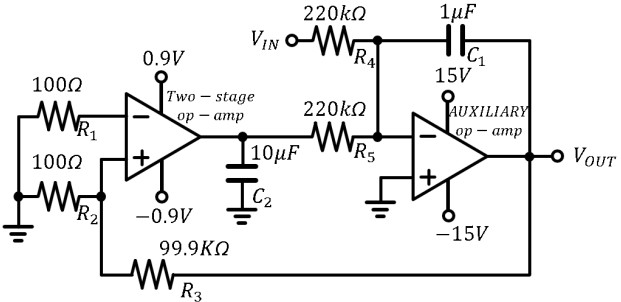

**Figure 18.** DC gain measurement circuit for two-stage op-amp.

Figure 19a shows the measurement results of the output voltage, and Figure 19b shows the result of averaging to remove the low-frequency noise of Figure 19a. The DC gain can be calculated from the slope in Figure 19b, which is 63.62 dB.

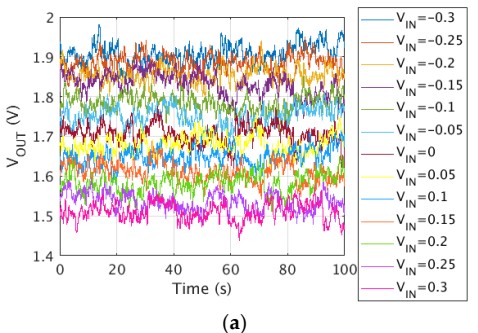
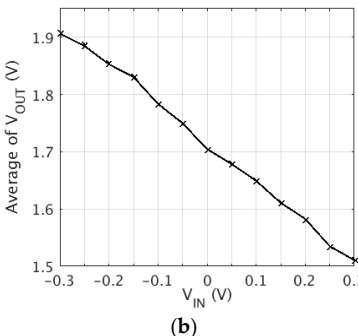

(**a**)        (**b**)

**Figure 19.** Measured output voltage of DC measurement circuit for two-stage op-amp. (**a**) Waveform of measured output voltage; (**b**) the mean of the output voltage.

The gain of the proposed op-amp is difficult to measure due to a large DC gain and flicker noise. A small resistor was added to the output stage to reduce the DC gain and measure it in order to predict the DC gain of the proposed op-amp. The measured DC gain was then compared with the DC gain of the conventional two-stage op-amp with the same output stage. Figure 20 shows the DC gain measurement circuit for the proposed op-amp with a small load resistance. $V_{IN}$ was changed from 0.1 V to 0.45 V at 50 mV intervals, and the $V_{OUT}$ was measured at 128 Sample/s for 500 s. Figure 21 shows the measurement result, with the DC gain measured at 70.46 dB.

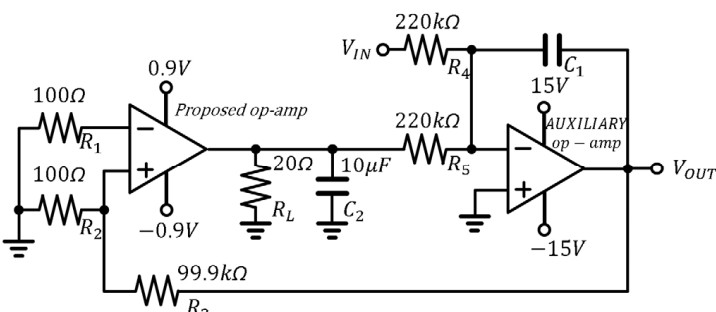

**Figure 20.** DC gain measurement circuit for the proposed op-amp with small load resistance.

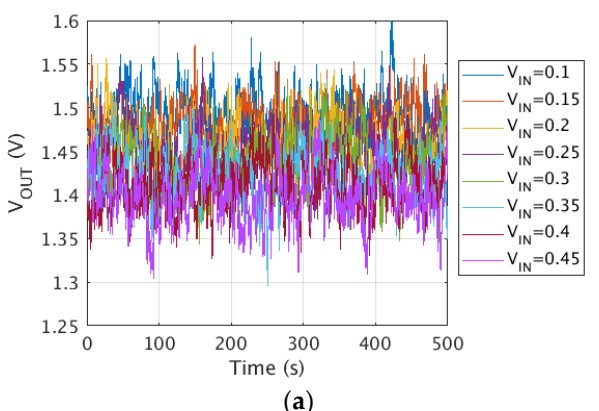
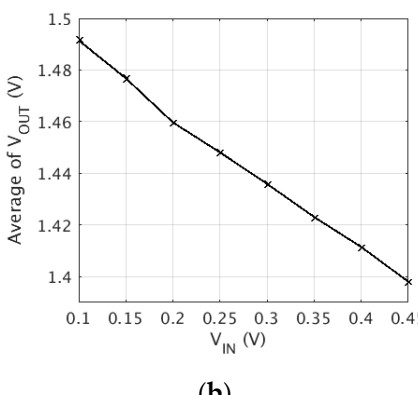

(**a**)        (**b**)

**Figure 21.** Measured output voltage of the DC measurement circuit for the proposed op-amp with small load resistance. (**a**) Waveform of the measured output voltage; (**b**) the mean of the output voltage.

Figure 22 shows the DC gain measurement circuit for the two-stage op-amp with small load resistance. $V_{IN}$ changed from 0.1 V to 0.45 V at 50 mV intervals, and the $V_{OUT}$ was measured at 1024 Sample/s for 100 s. Figure 23 shows the measurement result, and DC gain was measured at 34.25 dB.

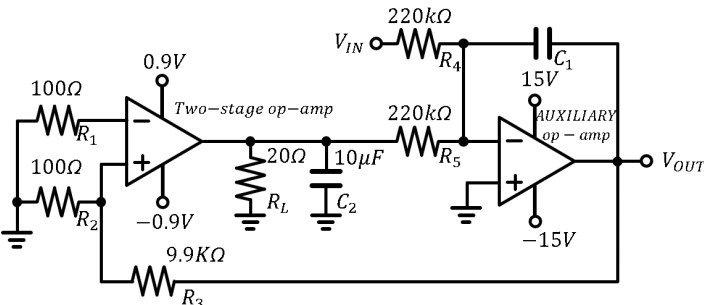

**Figure 22.** DC gain measurement circuit for the two-stage op-amp with small load resistance.

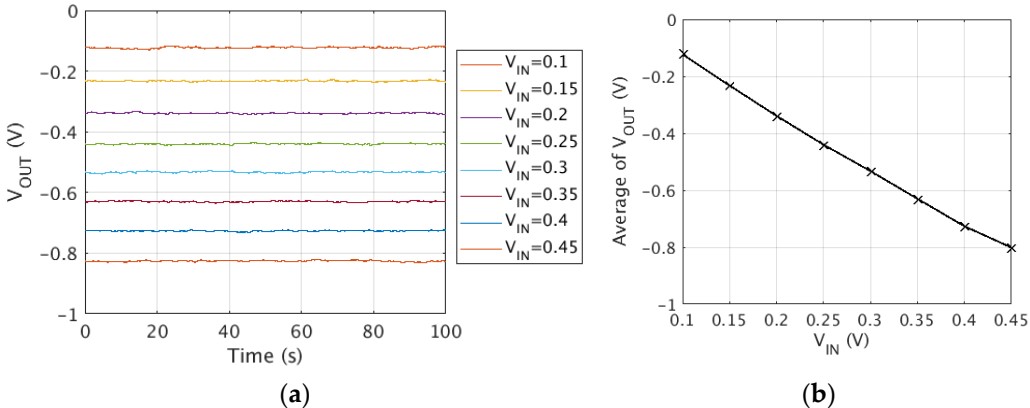

(**a**)  (**b**)

**Figure 23.** Measured output voltage of the DC measurement circuit for two-stage op-amp with small load resistance. (**a**) Waveform of measured output voltage; (**b**) the mean of output voltage.

The reduced DC gain of the proposed op-amp and two-stage op-amp ($A_{DC,3ST-withRL}$ and $A_{DC,2ST-withRL}$) can be expressed as Equations (43) and (44). The result of dividing the two equations is shown in Equation (45), which is equal to the ratio of the original DC gain.

$$A_{DC,3ST-withRL} = \frac{r_{oO} \parallel R_L}{r_{oO}} A_{DC,3ST-opamp} \tag{43}$$

$$A_{DC,2ST-withRL} = \frac{r_{oO} \parallel R_L}{r_{oO}} A_{DC,2ST-opamp} \tag{44}$$

$$\frac{A_{DC,3ST-withRL}}{A_{DC,2ST-withRL}} = \frac{A_{DC,3ST-opamp}}{A_{DC,2ST-opamp}} \tag{45}$$

The ratio of the two DC gains was measured at 36.21 dB, and the DC gain of the proposed op-amp was predicted to be 99.83 dB.

### 5.2. Unity–Gain Frequency and Phase Margin Measurement

The unity–gain frequency and phase margin were measured, with a sinewave of 50 mV amplitude supplied to the positive input. Figures 24 and 25 show the unity–gain frequency measurement results of the proposed op-amp and the two-stage op-amp. The unity–gain frequency and phase margin of the two-stage op-amp were 86.96 MHz and 54.8°, and those of the proposed op-amp were 86.96 MHz and 51.7°. Figure 26 shows the measured frequency response of the proposed op-amp and two-stage op-amp.

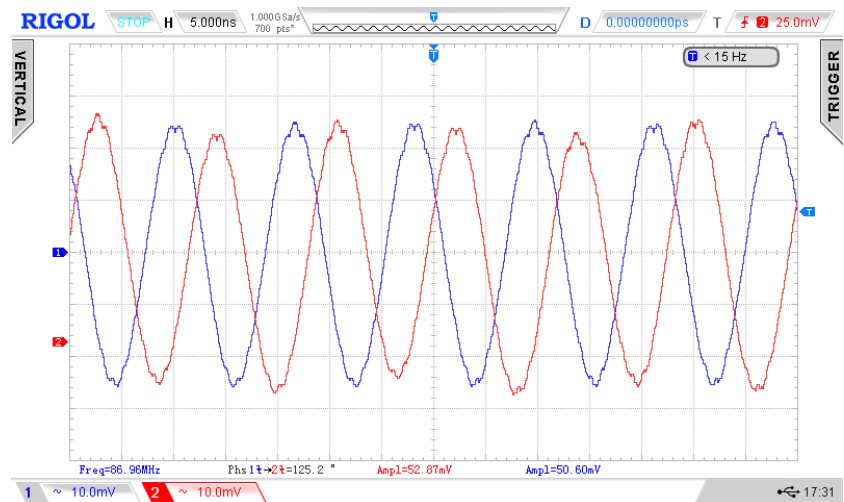

**Figure 24.** Unity–gain frequency measurement result of the two-stage op-amp.

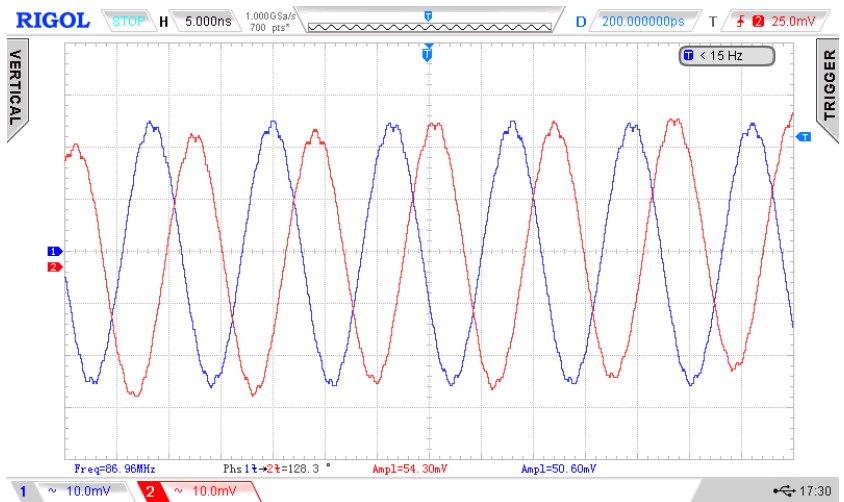

**Figure 25.** Unity–gain frequency measurement result of the proposed op-amp.

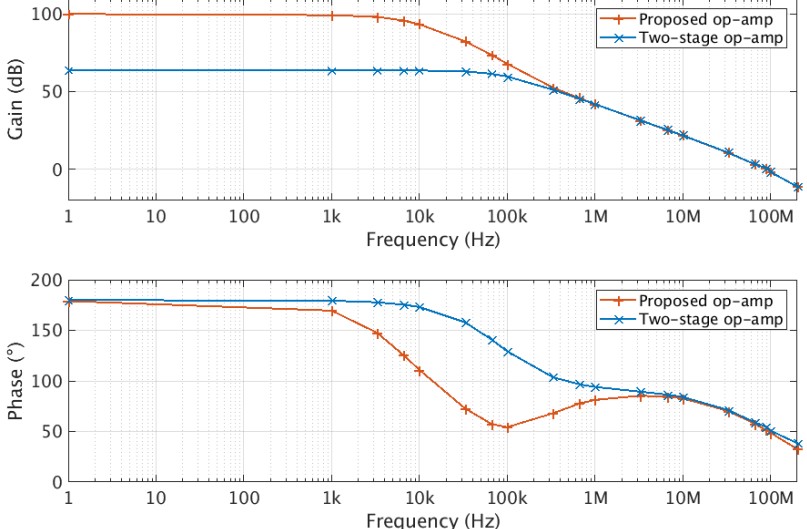

**Figure 26.** Measured frequency response of the proposed op-amp and two-stage op-amp.

### 5.3. Slew-Rate Measurement

Figures 27 and 28 show the measurement results of the slew-rate of the two-stage op-amp and the proposed op-amp. The input was a square wave of 100 mV amplitude and 10 MHz frequency, and the load capacitance was 90 pF. Both op-amps were measured to have a slew-rate of 32 MV/s.

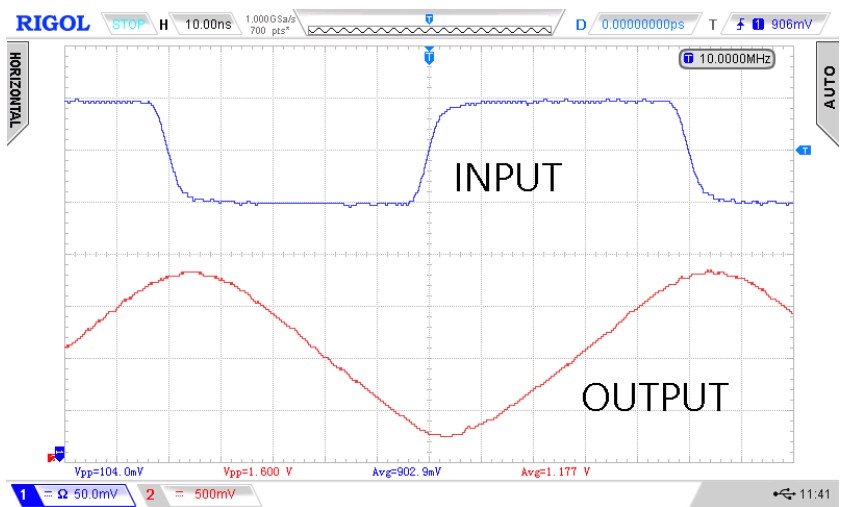

**Figure 27.** Measured frequency response of the proposed op-amp and two-stage op-amp.

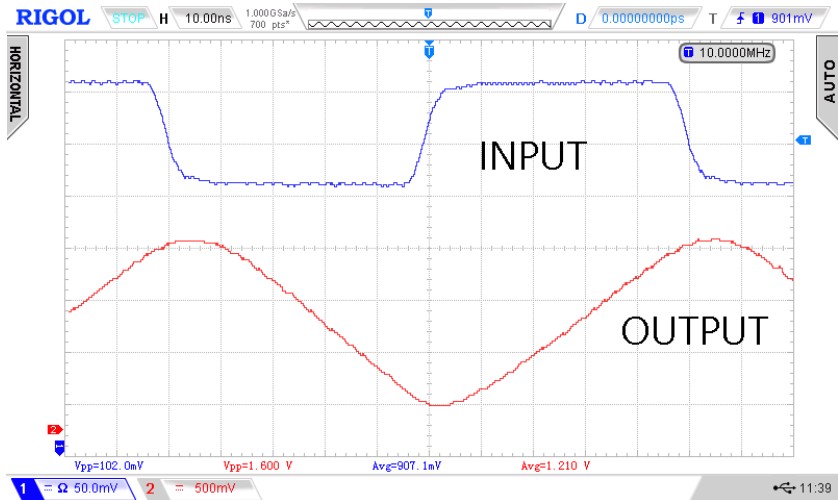

**Figure 28.** Measured frequency response of the proposed op-amp and two-stage op-amp.

The figures of merits (FOM) are expressed as follows [11]:

$$FOM_S = \frac{unity-gain\ frequency \times C_L}{Power} \tag{46}$$

$$FOM_L = \frac{SR \times C_L}{I_{VDD}} \tag{47}$$

$$IFOM_S = \frac{unity-gain\ frequency \times C_L}{Power} \tag{48}$$

$$IFOM_L = \frac{SR \times C_L}{I_{VDD}} \tag{49}$$

The $FOM_S$ and $FOM_L$ of the proposed op-amp were 359.4 $\left(\text{MHz·pF·mW}^{-1}\right)$ and 372 $\left(\text{V·pF·μs}^{-1}\text{·mW}^{-1}\right)$. $IFOM_S$ and $IFOM_L$ of the proposed op-amp were 647.0 $\left(\text{MHz·pF·mA}^{-1}\right)$ and 669.6 $\left(\text{V·pF·μs}^{-1}\text{·mA}^{-1}\right)$.

The summary of the measurement results of the two-stage op-amp and the proposed op-amp and comparison are shown in Table 3.

**Table 3.** Performance summary and comparison.

| Parameter | [6] | [10] | [11] | [12] | [13] | Two-Stage op-amp | This Work |
|---|---|---|---|---|---|---|---|
| Technology | 0.18 μm | 130 nm | 90 nm | 0.18 μm | 0.18 μm | 0.18 μm | 1.8 μm |
| Supply voltage (V) | 1.8 | 1.2 | 1.2 | 1.2 | 1.8 | 1.8 | 1.8 |
| Power (mW) | 0.85 | 0.1752 | 0.0204 | 1.8 | 0.86 | 7.742 | 7.742 |
| Core size (μm²) | - | 7000 | - | 1400 | 3038.5 | 6720 | 14,892 |
| $C_L$ (pF) | 5 | 12,000 | 500 | 5 | 5 | 32 | 32 |
| DC Gain (dB) | 105.5 | 107 | >100 | 65.5 | 82.7 | 63.62 | 99.83 |
| Unity–gain Frequency (MHz) | 231.77 | 1.18 | 4.65 | 146.9 | 88.7 | 86.96 | 86.96 |
| Phase margin (°) | 53 | 48.1 | 57 | 81.1 | 68.7 | 54.8 | 51.7 |
| Total compensation capacitance (pF) | 10.5 | 3.1 | 1.55 | 5 | 0.75 | 2.41 | 12.01 |
| FOM$_S$ (MHz·pF/mW) | 1214 | - | 113,970 | 548 | 516 | 359.4 | 359.4 |
| FOM$_L$ (V/μs·pF/mW) | 78 | - | 41,912 | - | 50 | 372 | 372 |
| I FOM$_S$ (MHz·pF/mA) | 2186 | 96,990 | 136,764 | 987 | 944 | 647 | 647 |
| I FOM$_L$ (V/μs·pF/mA) | 140 | 11,510 | 50,294 | - | 92 | 669.6 | 669.6 |

## 6. Conclusions

In this paper, a three-stage op-amp with frequency compensation using cascade zero was proposed. The area of the proposed op-amp was increased compared to the conventional two-stage op-amp due to the compensation capacitor, but the DC gain could be improved without any loss to the common-mode input range, output range, unity–gain frequency or power consumption. The proposed op-amp operated normally within the range of the process mismatch, as verified through the Monte Carlo simulation. This op-amp was implemented using 180 nm CMOS technology and measured to have a unity–gain frequency of 89.96 MHz, a phase margin of 51.7° and a DC gain of 99.83 dB, which was improved by 36.21 dB compared to the two-stage op-amp. This is suitable for pipeline analog-to-digital converters that require high gain and unity–gain frequency [14].

**Author Contributions:** Methodology, Y.J.; writing—original draft preparation, Y.J.; validation, Y.J.; writing—review and editing, Y.J., Y.S. and S.K.; project administration, S.C. All authors have read and agreed to the published version of the manuscript.

**Funding:** This work was supported by the Technology Innovation Program (20025097, Localization development of 16ch APD sensor module including ROIC for multichannel LiDAR sensor for vehicles) funded By the Ministry of Trade, Industry & Energy (MOTIE, Korea).

**Data Availability Statement:** The data presented in this study are available in the article.

**Acknowledgments:** This work was supported by the National Research Foundation (NRF), Korea, under project BK21 FOUR.

**Conflicts of Interest:** The authors declare no conflict of interest.

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
