# Peer review of "Three-Stage Operational Amplifier with Frequency Compensation Using Cascade Zero"

_electronics, doi:10.3390/electronics12112361_

Round 1
Reviewer 1 Report
The article describes an integrated three-stage amplifier compared to an already established two-stage solution.
The paper is well written and satisfactorily presents the background to the proposed frequency compensation.
I only looked superficially at the equations. Thus, I may have not seen a mistake.
Specific comments
When measuring the test circuit, why was the DC voltage amplification given so much attention but no direct comparison of the frequency response with the simulation made?
How many test circuits were measured?
Shouldn't in equation 41 DVin and Dvout be swapped?
Author Response
Thank you for your review.
I have written the response to the comments in a file
Please see the attachment.
Thank you.

Reviewer 2 Report
The manuscript presents a 3-stage opamp featuring a frequency compensation technique based on zeros related to the loading effect of the input of one stage on the output of the previous one ("cascade zeros").
- The topic is interesting, by the way the proposed technique seems to be similar to the one proposed in [1R] for zero-pole compensation in cascaded stage. Please comment on the differences and original aspects of the proposed approach.
- The results are supported by simulations and measurements, by the testing approach is non-standard. I recommend the authors to provide the measured Ad(f) transfer function in the frequency domain, not just the DC value and the UGB. Multi-dice measurements are also recommended.
- The manuscript is focused on the DC gain: it is quite straightforward that cascading more stages leads to a higher DC gain, once the closed-loop stability is properly achieved by the proposed frequency compensation approach. The effectiveness of the proposed technique should be discussed considering also GBW, phase margin and capacitive load in the theoretical assessment.
- The effectiveness of the proposed approach is demonstrated by comparison with a basic 2-stage amplifier, by the way more advanced compensation techniques have been proposed in recent years. A comparison table including recently proposed solutions should be provided.
[1R] M. A. Mohammed and G. W. Roberts, "Scalable Multi-Stage CMOS OTAs With a Wide CL-Drivability Range Using Low-Frequency Zeros," in IEEE Transactions on Circuits and Systems I: Regular Papers, vol. 70, no. 1, pp. 74-87, Jan. 2023, doi: 10.1109/TCSI.2022.3216201.
The English presentation is reasonable.
Author Response
Thank you for your review.
I have written the response to the comments in a file.
Please see the attachment.
Thank you.

Round 2
Reviewer 2 Report
The revised version is ok according to the reviewer.
The English quality is reasonable.